# Postpartum depressive symptoms following implementation of the 10 steps to successful breastfeeding program in Kinshasa, Democratic Republic of Congo: A cohort study

Robert A. Agler[1,2], Paul N. Zivich[3,4], Bienvenu Kawende[5], Frieda Behets[3], Marcel Yotebieng[1,6]*

1 Division of Epidemiology, College of Public Health, The Ohio State University, Ohio, United States of America, 2 Department of Psychology, The Ohio State University, Ohio, United States of America, 3 Department of Epidemiology, University of North Carolina at Chapel Hill, North Carolina, United States of America, 4 Carolina Population Center, University of North Carolina at Chapel Hill, North Carolina, United States of America, 5 School of Public Health, The University of Kinshasa, Kinshasa, Democratic Republic of Congo, 6 Division of General Internal Medicine, Albert Einstein College of Medicine, New York, United States of America

* Marcel.yotebieng@einsteinmed.org, myotebieng@gmail.com

**Data Availability Statement:** The data underlying the results presented in the study is provided in the

## Abstract

### Background

Social support and relevant skills training can reduce the risk of postpartum depression (PPD) by reducing the impact of stressors. The 10-step program to encourage exclusive breastfeeding that forms the basis of the Baby-Friendly Hospital Initiative (BFHI) provides both, suggesting it may lessen depressive symptoms directly or by reducing difficulties associated with infant feeding. Our objective was to quantify the association of implementing Steps 1–9 or Steps 1–10 on postpartum depressive symptoms and test whether this association was mediated by breastfeeding difficulties.

### Methods and findings

We used data from a breastfeeding promotion trial of all women who gave birth to a healthy singleton between May 24 and August 25, 2012 in 1 of the 6 facilities comparing different BFHI implementations (Steps 1–9, Steps 1–10) to the standard of care (SOC) randomized by facility in Kinshasa, Democratic Republic of Congo. Depressive symptoms, a non-registered trial outcome, was assessed at 14 weeks via the Edinburgh Postnatal Depression Scale (EPDS). Inverse probability weighting (IPW) was used to estimate the association of BFHI implementations on depressive symptoms and the controlled direct association through breastfeeding difficulties at 10 weeks postpartum.

A total of 903 mother–infant pairs were included in the analysis. Most women enrolled had previously given birth (76%) and exclusively breastfed at 10 weeks (55%). The median age was 27 (interquartile range (IQR): 23, 32 years). The proportion of women reporting breastfeeding difficulties at week 10 was higher in both Steps 1–9 (75%) and Steps 1–10

supporting data file (S1 Data) as well as the SAS code (S1 SAS Code).

**Funding:** This original trial was supported by a grant from the Bill & Melinda Gates Foundation to FHI 360, through the Alive and Thrive Small Grants Programme (University of California, Davis, CA, USA) via a subaward to the University of North Carolina (OH, USA; subagreement 09-000076-AT11-123-UNC-DRC). PNZ is supported by National Institute of Child Health and Human Development T32-HD091058. RAA is supported by the National Institute of Health (U01AI096299). MY is partially supported by the National Institute of Health (U01AI096299 and 1R01H087993). The funders had no role in study design, data collection, data collection, data analysis and interpretation, preparation of the manuscript, or decision to submit.

**Competing interests:** The authors have declared that no competing interests exist.

**Abbreviations:** BFHI, Baby-Friendly Hospital Initiative; CI, confidence interval; CL, confidence level; EPDS, Edinburgh Postnatal Depression Scale; IPCW, inverse probability of censoring weight; IPTW, inverse probability of treatment weight; IPW, inverse probability weight; IQR, interquartile range; IRB, Institutional Review Board; PD, prevalence difference; PPD, postpartum depression; SES, socioeconomic status; SOC, standard of care; STROBE, Strengthening the Reporting of Observational Studies in Epidemiology.

(91%) relative to the SOC (67%). However, the number of reported difficulties was similar between Steps 1–9 (median: 2; IQR: 0, 3) and SOC (2; IQR: 0, 3), with slightly more in Steps 1–10 (2; IQR: 1, 3). The prevalence of symptoms consistent with probable depression (EPDS score >13) was 18% for SOC, 11% for Steps 1–9 (prevalence difference [PD] = −0.08; 95% confidence interval (CI): −0.14 to −0.01, $p = 0.019$), and 8% for Steps 1–10 (PD = −0.11, −0.16 to −0.05; $p < 0.001$). We found mediation by breastfeeding difficulties. In the presence of any difficulties, the PD was reduced for both Steps 1–9 (−0.15; 95% confidence level (CL): −0.25, −0.06; $p < 0.01$) and Steps 1–10 (−0.16; 95% CL: −0.25, −0.06; $p < 0.01$). If no breastfeeding difficulties occurred in the population, there was no difference in the prevalence of probable depression for Steps 1–9 (0.21; 95% CL: −0.24, 0.66; $p = 0.365$) and Steps 1–10 (−0.03; 95% CL: −0.19, 0.13; $p = 0.735$). However, a limitation of the study is that the results are based on 2 hospitals randomized to each group.

## Conclusions

In conclusion, in this cohort, the implementation of the BFHI steps was associated with a reduction in depressive symptoms in the groups implementing BFHI Steps 1–9 or 1–10 relative to the SOC, with the implementation of Steps 1–10 associated with the largest decrease. Specifically, the reduction in depressive symptoms was observed for women reporting breastfeeding difficulties. PPD has a negative impact on the mother, her partner, and the baby, with long-lasting consequences. This additional benefit of BFHI steps suggests that renewed effort to scale its implementation globally may be beneficial to mitigate the negative impacts of PPD on the mother, her partner, and the baby.

## Trial registration

ClinicalTrials.gov NCT01428232

### Author summary

#### Why was this study done?

- Depression is common in the postpartum period. Untreated, postpartum depression (PPD) has substantial negative impact on the mother, her partner, and the baby, with long-lasting consequences.

- Social support and relevant skills training can reduce the risk of PPD by reducing the impact of stressors.

- The 10-step program to encourage exclusive breastfeeding that forms the basis of the Baby-Friendly Hospital Initiative (BFHI) provides both, but its effect on depression symptoms in postpartum period has not been evaluated.

### What did the researchers do and find?

- We found that depressive symptoms were reduced in the groups implementing BFHI Steps 1–9 or 1–10 relative to the standard of care (SOC), with the implementation of Steps 1–10 associated with the largest decrease.

- The reduction in depressive symptoms associated with the implementation of BHFI Steps 1–9 or Steps 1–10 was observed for women reporting breastfeeding difficulties, but not for women reporting no difficulties.

### What do these findings mean?

- Our results suggest that the implementation of the BFHI was associated with a reduction in maternal depressive symptoms and that breastfeeding difficulties play a role in this relationship.

## Introduction

The postpartum period is associated with numerous physical, emotional, social, and financial stressors, many of which increase the risk of postpartum depression (PPD) [1,2]. Infant feeding difficulties are a particularly common source of stress, with some 60% to 90% of women reporting difficulties with breastfeeding (e.g., pain, proper infant positioning, etc.), leading to an increased risk of PPD, and uncertainty and ambivalence about different methods of feeding [3,4]. The 10 steps to successful breastfeeding, i.e., the basis of the Baby-Friendly Hospital Initiative (BFHI; Table 1), helps to address many issues tied to breastfeeding difficulties, as recent mothers are given a combination of knowledge, skills training, encouragement, and social support to facilitate optimal feeding practices [5]. This reduction in stress provides a pathway through which the BFHI may reduce symptoms of PPD, but this possibility has not been assessed.

**Table 1. Ten steps that serve as the basis for the BFHI.**

| Step | Definition |
|---|---|
| 1 | Having a written breastfeeding policy that is routinely communicated to all healthcare staff |
| 2 | Training all healthcare staff in skills necessary to implement this policy |
| 3 | Informing all pregnant women about the benefits and management of breastfeeding |
| 4 | Helping mothers to initiate breastfeeding within 30 minutes of birth |
| 5 | Showing mothers how to breastfeed and maintain lactation, even if they are separated from their infants |
| 6 | Giving newborn infants no food or drink other than breastmilk, unless medically indicated, and not accepting free or low-cost breastmilk substitutes, feeding bottles, or teats |
| 7 | Allowing mothers and infants to remain together 24 hours per day |
| 8 | Encouraging breastfeeding on demand |
| 9 | Giving no artificial teats or pacifiers to breastfeeding infants |
| 10 | Fostering the establishment of breastfeeding support groups and referring mothers to them on discharge from a hospital or clinic |

BFHI, Baby-Friendly Hospital Initiative.

To further elaborate, parallels can be drawn between the BFHI and psychotherapeutic interventions, although we do not claim that the BFHI can replace therapy. Both identify stressors, reframe them in a less stressful way, teach how to manage the stressors, and provide encouragement and social support to motivate people to effectively deal with stressors [5,6]. For psychotherapy, the result is that patients express greater motivation, successful goal achievement, and are less discouraged by stressors. Mirroring this pattern, mothers that receive breastfeeding interventions express greater intention to breastfeed, higher rates of exclusive breastfeeding, and are more likely to persist when encountering breastfeeding difficulties [7,8]. Both also provide an opportunity to openly discuss and address potential sources of shame and uncertainty. Psychiatric illnesses and infant feeding practices are also both potentially stigmatizing, as both are often heavily moralized [9,10]. Shame and threat of social rejection are risk factors for the development of psychiatric disorders and a barrier to their management [9]. Mothers regularly report worries of judgment and shaming, as well as instances of outright shaming, that can make mothers feel isolated or inadequate [10]. Open, honest, and judgment-free discussions of shameful topics helps reduce any feelings of shame they may cause [11]. Additionally, simply being uncertain and afraid of the unknown (e.g., concerns about harm to the child based on feeding method) is a risk factor for the development of psychiatric disorders [12].

The community breastfeeding support groups (Step 10) provide an ongoing source of support and knowledge beyond what hospital staff provides at follow-up visits. Support group meetings also last far longer than the brief interactions possible at a hospital, allowing for deeper discussion of ongoing or recent issues and for further provision of both objective (e.g., skills and knowledge) and subjective social support (e.g., encouragement and conveying that the mothers' problems are neither unique nor indicative of moral character). Additionally, while social support from relative strangers, such as hospital staff, is helpful, social support provided by familiar, closer people provides stronger benefits [13,14]. Further, the opportunity to help others may also reduce stress, as prosocial behaviors result in psychological benefits for the giver [15]. In sum, the BFHI provides mothers with a variety of sources and aid to help with common postpartum stressors that have been shown to increase the risk of PPD. Further, by increasing early initiation (within 1 hour of birth) and prolonged exclusive breastfeeding [16], BFHI interventions are likely associated with increased levels of oxytocin among mothers. Higher levels of oxytocin are associated with lower reported breastfeeding difficulties and improved maternal mood [17,18].

The 10 steps can thus be broken down into 3 groups. Steps 1 and 2 are meant to provide the overall structure of breastfeeding support and care, establishing facility-level policies and guidelines, as well as their implementation. Steps 3–9 provide in-hospital care, with hospital staff encouraging the initiation of breastfeeding, as well as providing information about its benefits to mothers. Step 10 is about community-related care, with hospital staff providing referrals to community-based breastfeeding support groups that can serve as sources of additional support and care for breastfeeding. Combined with the effects of exclusive breastfeeding, which also reduces the risk of PPD [19], the implementation of the 10 steps is expected to reduce depressive symptoms. However, to the authors' knowledge, the effects of the BFHI on postpartum depressive symptoms have not been previously investigated.

Our objective was to quantify the association of implementing Steps 1–9 or Steps 1–10 on postpartum depressive symptoms. Specifically, we expected that Steps 1–9 (in-hospital care) would be associated with a decrease in depressive symptoms and that when combined with Step 10 (community support groups), the decrease would be larger. Reflecting the hypothesized reduction in stress, we also expected that breastfeeding difficulties would mediate the association of the BFHI on depressive symptoms.

## Materials and methods

Data come from a cluster randomized trial of breastfeeding promotion in Kinshasa, Democratic Republic of Congo (NCT01428232) [16]. Depression was not included in registered trial outcomes, and although we have anticipated looking at the effect of the interventions on the prevalence of depressive symptoms, the plan for this analysis was not included in the study protocol (S1 Protocol) and was developed after the trial was funded. The primary purpose of the original study was to investigate whether a partial implementation without Step 10 would be effective in promoting exclusive breastfeeding and whether training nurses to provide additional support to breastfeeding mothers could be used as an alternative implementation of Step 10. Analyses of the primary and secondary outcomes of the trial showed that the implementation of basic training in BFHI Steps 1–9 had no additional effect on initiation of breastfeeding but significantly increased exclusive breastfeeding at 6 months of age and significantly reduced incidence of mild and severe episodes of diarrhea and respiratory infection in the first 6 months of life. Additional support based on the same training materials and locally available breastfeeding support materials, offered during well-child visits (i.e., Step 10), appeared to lessen this effect [16,20].

Six clinics grouped in pair based on their similar workload, number of deliveries, and proportion of mothers returning for their 1 week postpartum visit were randomized to either standard of care (SOC), Steps 1–9 group, or Steps 1–10 group (Fig 1). To inform the selection and pairing, study staff visited 44 facilities in Kinshasa between October and November 2011 and collected information about factors that might affect the quality of care provided in each facility. This selection process was intended to account for differences in important variables, such as utilization of services and socioeconomic status (SES), and to minimize the overall heterogeneity of clinic characteristics. Clinics were randomized rather than mother–infant pairs to minimize cross-group contamination and to better reflect the implementation of the BFHI in practice [16]. Randomization was performed by a statistician not involved in field activities. Staff providing the intervention could not be blinded, but independent interviewers and participants were blinded. The implementation of the BFHI was done by healthcare staff trained with WHO/UNICEF BFHI course. Culturally appropriate informational flyers in official (French) and local (Lingala) languages were provided at facilities randomized to Steps 1–10. The implementation of Steps 1–9 was assessed at the end of the study using the hospital self-appraisal questionnaire, and each of the clinics randomized to intervention groups met at least 80% of the global criteria for each step. In facilities randomized to SOC, besides study briefing prior to randomization, nothing else was provided aside from routine care. The SOC is described in detail elsewhere [21]. The average total number of deliveries in the year preceding the study was comparable across groups: 1,375 in the SOC compared to 1,520 and 1,291 in the Steps 1–9 and 1–10 groups, respectively. However, the average total number of healthcare personnel (physicians, nurses, and nurse assistant) was lower in the Steps 1–9 group (12) as compared to SOC (16) and Steps 1–10 (19) groups. Consequently, the average workload total (deliveries/total personnel) was substantially higher in Steps 1–9 group (144) relative to SOC (90) and Steps 1–10 (83) groups.

All women who gave birth to a healthy singleton in selected facilities between May 24 and August 25, 2012 were eligible to be enrolled in the study if they intended to attend well-baby clinic visits in the respective facilities. Women were enrolled within 2 days of delivery and follow-up at 6, 10, 14, 18, and 24 weeks postpartum.

### Ethics statement

The original study design and implementation was approved by the University of North Carolina at Chapel Hill Institutional Review Board (IRB) and the Kinshasa School of Public Health

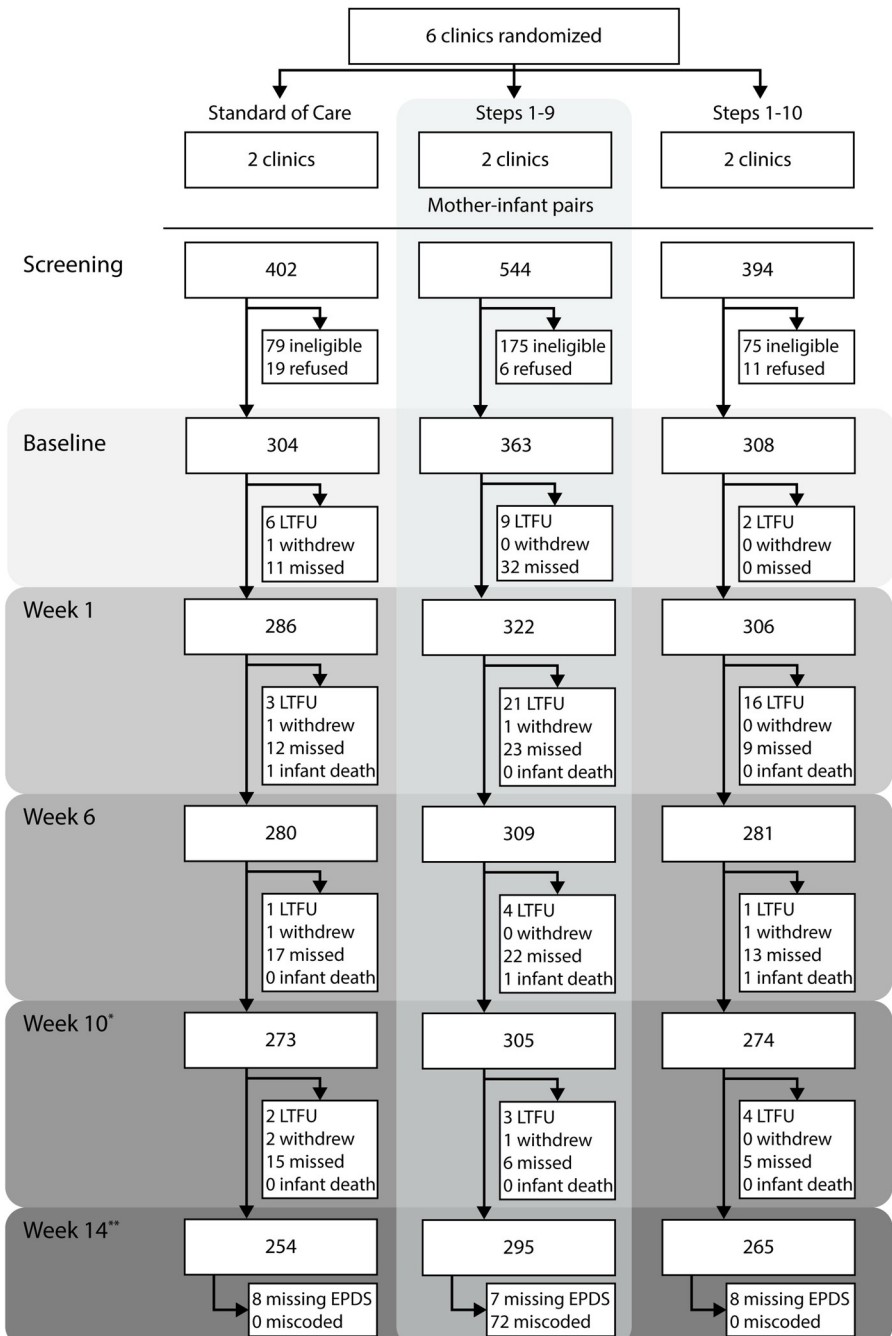

**Fig 1. Flow diagram of follow-up of mother–infant pairs.** Mothers who missed the week 10 visit were considered censored at week 14 to ensure that censoring was monotonic. One reviewer in the Steps 1–9 group was believed to have incorrectly recorded the EPDS scores for participants. EPDS scores of participants assessed by this 1 staff member were set as missing. * Reported breastfeeding difficulties were assessed at week 10. ** Depressive symptoms were assessed as week 14 via the EPDS. EPDS, Edinburgh Postnatal Depression Scale; LTFU, lost to follow-up.

Ethical Committee (see attached funded proposal in the Supporting information: S1 Protocol). All participants provided written informed consent. This analysis was approved by the Ohio State University IRB as secondary analysis of de-identified data.

## Measures

At enrollment, demographics and obstetrical history were collected. Demographic information included age (continuous), education (primary; secondary or higher), marital status (married/live-in boyfriend; never married/separated/divorced), previous children (yes; no), prior miscarriage/abortion/stillbirth (yes; no), not wanting additional children (yes; no), at least 4 antenatal clinic visits before delivery (yes; no), and prior experiences of domestic violence (yes; no). Home ownership (yes; no) and a flush toilet facility (yes; no) were used as indicators of relative SES. The woman's experience at the clinic was measured as being given their baby right after delivery (yes; no). Exclusive breastfeeding (yes; no) was assessed with retrospective recall for the previous week and defined as no food or water sources other than breastmilk at each follow-up visit. Proportion of feedings (continuous) that were breastfeeding was calculated as the reported number of breastfeeding divided by the total number of reported feedings for the prior 7 days. In predefined sensitivity analysis, the proportions of feedings that were breastfeeding within the past week at week 10 were considered instead of exclusive breastfeeding up to week 10. At each study visit, difficulties experienced with breastfeeding were defined as any self-reported problems with breastfeeding since the week 6 visit. The number of difficulties breastfeeding was calculated as a sum of all reported problems. For this analysis, only exclusive breastfeeding and difficulties breastfeeding at week 10, the visit prior to week 14 when depression symptoms were assessed are considered.

Depression symptoms were assessed at 14 weeks postpartum with the Edinburgh Postnatal Depression Scale (EPDS) [22]. The EPDS is a short 10-item screening tool to assess PPD symptoms that has been shown to have good sensitivity and specificity across many settings [23]. The original English scale was translated to both French and Lingala by translators fluent in all 3 languages and back translated to ensure fidelity. Scores of 13 or higher were considered to reflect probable depression [22]. To ensure the monotonicity of loss to follow-up, women who missed their week 10 visit were considered as censored and had their EPDS score set to missing. One study staff member had systematically different EPDS scores and was believed to have incorrectly recorded the EPDS scores for participants. Specifically, the items that were reverse-scored had systematically lower values than other reviewers. The participants at a clinic the Steps 1–9 arm assessed by this 1 staff member ($n$ = 72) had their EPDS scores set to missing.

## Statistical analysis

To estimate the 10-week prevalence difference (PD) in any breastfeeding difficulties, 10-week difference in number of breastfeeding difficulties, 14-week PD of postpartum probable depression (EPDS scores ≥13), and 14-week difference in EPDS scores, we used inverse probability of treatment weights (IPTWs) to account for potential baseline confounders. IPTWs account for potential confounders of the intervention and depression symptoms by creating a weighted population where observed confounders are no longer imbalanced between arms of the intervention [24]. Stabilized IPTWs were calculated as the unconditional probability of the intervention arm divided by the probability of the intervention arm conditional on observed confounders (see Supporting information, S1 IPW Calculation for details on stabilized IPTW calculation). Observed confounders included age, marital status, education, previous miscarriage, parity, whether the pregnancy was wanted, indicators of SES, antenatal care visits, and previous domestic violence. Probabilities for the intervention arms were generated from a multinomial logistic regression model. Age was modeled using restricted quadratic splines with knots at 20, 30, and 40.

To relax the assumption that loss to follow-up was non-informative, we used inverse probability of censoring weights (IPCWs) to account for potentially informative censoring by observed variables [25]. Stabilized IPCWs were calculated conditional on factors believed to be

related to loss to follow-up. IPCWs for weeks 10 and 14 were constructed using a logistic regression model predicting censoring as a function of trial intervention arm, number of breastfeeding difficulties, demographics, and clinic experience.

Inverse probability weights (IPWs) were additionally used to estimate the controlled direct associations with probable depression at 14 weeks postpartum, where the controlled direct association corresponds to the change in probable depression if breastfeeding difficulties had uniformly been set to the same level in the population and BFHI steps implementations were compared [26]. Stabilized IPWs for mediation were calculated as the probability of the mediator conditional on the intervention arm divided by the probability of the mediator conditional on the intervention arm and observed confounders. For mediation by number of breastfeeding difficulties, a Poisson model was used to calculate probabilities. Since no women in the SOC group reported more than 4 difficulties, weights and models were estimated only for women reporting 4 or fewer difficulties. A logistic model was used for any breastfeeding difficulties as the mediator. IPWs were conditional on the same confounders as the intervention IPW and further included exclusive breastfeeding at 10 weeks and whether their baby was given right after delivery. Further details on construction of all IPW are provided in the Supporting information (S1 IPW Calculation).

Estimated IPWs were used to fit corresponding marginal structural models. For count variables (EPDS scores and number of breastfeeding difficulties), a linear Poisson marginal structural model was used. A linear binomial model was used for binary outcomes (postpartum probable depression and any difficulties breastfeeding). Saturated marginal structural models were fit for the average total association by intervention arms and the controlled direct association for mediation by any difficulties breastfeeding at 10 weeks. For mediation by the number of breastfeeding difficulties at week 10, the number of breastfeeding difficulties was modeled using a quadratic term. Moreover, 95% confidence intervals (CIs) and *p*-values for estimated PD were calculated with robust variance estimators, which provides conservative coverage [27]. All analyses were conducted with SAS 9.4 (Cary, North Carolina, United States of America). This study is reported as per the Strengthening the Reporting of Observational Studies in Epidemiology (STROBE) guidelines (S1 STROBE Checklist).

## Results

Fig 1 presents the flow diagram of mother–infant pairs through the eligible follow-up period for the presented analyses. In total, 975 mother–infant pairs enrolled in the study, and 718 were considered as uncensored at week 14 postpartum (Table 2). The prevalence of probable depression at 14 weeks postpartum was 18% for SOC, 11% for Steps 1–9, and 8% for Steps 1–10. Median EPDS scores were 8 (interquartile range (IQR): 3, 11) for the SOC, 6 (IQR: 2, 9) for Steps 1–9, and 3 (IQR: 0, 8) for Steps 1–10. More women in the Steps 1–9 (75%) and Steps 1–10 (91%) groups reported difficulties breastfeeding at week 10 compared to SOC (67%). The most commonly reported difficulties of breastfeeding at week 10 were the following: breasts were overfull, the baby nursed too often, and breasts leaked too much (Table 3).

BFHI intervention implementations were associated with reduced EPDS scores and the prevalence of depressive symptoms when compared to SOC (Table 4). Relative to the SOC, both Steps 1–9 (0.42; 95% confidence level (CL): 0.15, 0.70; *p* < 0.01) and Steps 1–10 (0.48; 95% CL: 0.22, 0.74; *p* < 0.01) had had a greater number of reported breastfeeding difficulties for IPTW with IPCW. Additionally, the prevalence of any reported difficulties was higher for Steps 1–9 (0.08; 95% CL: −0.01, 0.17; *p* = 0.06) and Steps 1–10 (0.22; 95% CL: 0.14, 0.30; *p* < 0.01) as well.

Among both intervention groups, the associations of the BFHI with probable depression were mediated through the prevalence of any breastfeeding difficulties (Table 5). In the

**Table 2. Characteristics of women at study enrollment by intervention group (n = 975).**

| | SOC (n = 304) | Steps 1–9 (n = 363) | Steps 1–10 (n = 308) |
|---|---|---|---|
| EPDS score at 14 weeks (IQR) | 8 (3, 11) | 6 (2, 9) | 3 (0, 8) |
| Missing | 58 | 148 | 51 |
| Probable depression at 14 weeks* | 45 (18%) | 23 (11%) | 20 (8%) |
| Any breastfeeding difficulties† | 183 (67%) | 229 (75%) | 249 (91%) |
| Missing | 31 | 59 | 34 |
| Number of breastfeeding difficulties† (IQR) | 2 (0, 3) | 2 (0, 3) | 2 (1, 3) |
| Missing | 31 | 59 | 34 |
| Exclusive breastfeeding up to week 10 | 117 (38%) | 237 (65%) | 178 (58%) |
| Proportion of feedings that were with breastmilk | 0.95 (0.83, 1) | 1 (1, 1) | 1 (0.91, 1) |
| Missing | 31 | 62 | 34 |
| Age (IQR) | 28 (24, 33) | 26 (23, 32) | 26 (22, 31) |
| Missing | 9 | 3 | 4 |
| Marital status | | | |
| Married/live-in boyfriend | 269 (88%) | 322 (89%) | 256 (83%) |
| Separated/divorced/never married | 35 (12%) | 41 (11%) | 51 (17%) |
| At least 4 ANC visits | 213 (70%) | 169 (47%) | 129 (42%) |
| Baby given right after birth | 58 (19%) | 266 (74%) | 114 (38%) |
| Missing | 0 | 2 | 5 |
| Previous miscarriage/abortion/stillbirth | 91 (30%) | 118 (33%) | 99 (32%) |
| Education | | | |
| Primary or less | 30 (10%) | 67 (19%) | 36 (12%) |
| Secondary or higher | 263 (90%) | 285 (81%) | 269 (88%) |
| Missing | 11 | 11 | 3 |
| Own house | 146 (48%) | 130 (36%) | 139 (45%) |
| Missing | 0 | 3 | 0 |
| Flush toilet | 81 (26%) | 179 (49%) | 158 (51%) |
| Wanted child at conception | 145 (48%) | 187 (52%) | 103 (33%) |
| Missing | 1 | 0 | 0 |
| Domestic violence | 80 (26%) | 61 (17%) | 98 (32%) |
| Parity‡ | 223 (73%) | 285 (79%) | 229 (74%) |

ANC, antenatal care; BFHI, Baby-Friendly Hospital Initiative; EPDS, Edinburgh Postnatal Depression Scale; IQR, interquartile range; SOC, standard of care.

Steps 1–9 consisted of mother–infant pairs attending clinics implementing Steps 1 through 9 of the BFHI. Steps 1–10 implemented all Steps 1 through 10.

* Probable depression at 14 weeks postpartum was defined as an EPDS of 13 or greater.

† Difficulties were reported at 10 weeks postpartum.

‡ Parity is defined as having a previous child.

presence of difficulties, EPDS scores were decreased in Step 1–9 (−3.65; 95% CL: −4.80, −2.50; $p < 0.01$) and Steps 1–10 (−5.21; 95% CL: −6.33, −4.10; $p < 0.01$) compared to SOC. A similar relationship occurred for probable depression prevalence for Steps 1–9 (−0.15; 95% CL: −0.25, −0.06; $p < 0.01$) and Steps 1–10 (−0.16; 95% CL: −0.25, −0.06; $p < 0.01$). When no breastfeeding difficulties occurred, there was a positive but imprecise association of both interventions with EPDS scores or probable depression, 2.92; 95% CL: −7.04, 12.89; $p = 0.57$ for Steps 1–9 and 1.46; 95% CL: −1.32, 4.23; $p = 0.30$ for Steps 1–10 (Table 5). There is insufficient indication of increased EPDS scores for either intervention if no breastfeeding difficulties was held constant. Mediation remained apparent when considering the number of breastfeeding difficulties instead. At 0 reported difficulties, Steps 1–9 and Steps 1–10 had little to no effect (Supporting

**Table 3. Reported breastfeeding difficulties at 10 weeks postpartum by intervention group (*n* = 851).**

|  | SOC (*n* = 273) | Steps 1–9 (*n* = 304) | Steps 1–10 (*n* = 274) |
|---|---|---|---|
| Breasts leaked too much | 111 (41%) | 90 (30%) | 112 (41%) |
| Breasts were overfull | 157 (58%) | 193 (63%) | 171 (62%) |
| Breast infection | 1 (0%) | 1 (0%) | 0 |
| Yeast infection of breast | 0 | 0 | 0 |
| Nipples were sore, cracked, and bleeding | 8 (3%) | 9 (3%) | 16 (6%) |
| Clogged milk duct | 0 | 0 | 0 |
| Trouble getting milk flow to start | 1 (0%) | 13 (4%) | 2 (1%) |
| Not enough milk | 11 (4%) | 17 (6%) | 19 (7%) |
| Too long for milk to come in | 0 | 43 (14%) | 20 (7%) |
| Baby choked | 0 | 15 (5%) | 17 (6%) |
| Baby would not wake up to nurse regularly enough | 3 (1%) | 64 (21%) | 35 (13%) |
| Baby not interested in nursing | 18 (7%) | 23 (8%) | 21 (8%) |
| Baby nursed too often | 157 (58%) | 164 (54%) | 161 (59%) |
| Baby had trouble sucking or latching on | 0 | 0 | 0 |
| Baby did not gain enough weight | 0 | 0 | 5 (2%) |
| Other problem | 0 | 0 | 26 (9%) |

BFHI, Baby-Friendly Hospital Initiative; SOC, standard of care.

Steps 1–9 consisted of mother–infant pairs attending clinics implementing Steps 1 through 9 of the BFHI. Steps 1–10 implemented all Steps 1 through 10.

information: S1 Sensitivity Analysis, Table 3.1). In general, as the number of difficulties increased, the strength of the protective association between BFHI and EPDS score or probable depression increased until 3 difficulties. At 4 reported difficulties, the strength of the protective association flattened or decreased (Figs 2 and 3).

## Sensitivity analysis

To assess the sensitivity of the finding to how breastfeeding was defined in the IPW mediation model, we repeated the analysis but with the proportions of feedings that were breastfeeding within the past week at week 10; instead of exclusive breastfeeding up to week 10. As shown in the Supporting information (S1 Sensitivity Analysis), results were similar to the main results. The controlled direct association for both EPDS scores and probable depression reversed directions under no breastfeeding difficulties (Supporting information: S1 Sensitivity Analysis, Table 3.1). These results for mediation for Steps 1–9 are in contrast to Table 5 where increases in EPDS scores and probable depression were seen for Steps 1–9 under no difficulties breast-feeding. However, all these results were imprecise and near the null value. The estimated controlled direct association for Steps 1–9 under no difficulties breastfeeding were dependent on how breastfeeding at week 10 was defined. Together, these results indicate that the association between Steps 1–9 and EPDS is null or relatively small if the population had been held constant at no breastfeeding difficulties at week 10. The remainder of mediation results were consistent across the different approaches to operationalizing breastfeeding at week 10.

## Discussion

We hypothesized that the BFHI steps may be understood as an intervention that helps mothers deal with common postpartum stressors, reducing depressive symptoms. Our results provide evidence that the BFHI is associated with reduce postpartum depressive symptoms compared

**Table 4. Intervention group association with EPDS and breastfeeding difficulties.**

| | Steps 1–9 | | | Steps 1–10 | | |
|---|---|---|---|---|---|---|
| | Point estimate | 95% CI | p | Point estimate | 95% CI | p |
| EPDS score* | | | | | | |
| Unadjusted | −1.51 | (−2.50, −0.51) | 0.003 | −3.05 | (−3.99, −2.11) | <0.001 |
| IPTW‡ | −1.18 | (−2.32, −0.04) | 0.043 | −2.85 | (−3.91, −1.79) | <0.001 |
| IPTW with IPCW# | −1.55 | (−2.70, −0.40) | 0.008 | −3.11 | (−4.18, −2.03) | <0.001 |
| Probable depression† | | | | | | |
| Unadjusted | −0.08 | (−0.14, −0.01) | 0.019 | −0.11 | (−0.16, −0.05) | <0.001 |
| IPTW‡ | −0.07 | (−0.15, 0.00) | 0.059 | −0.10 | (−0.17, −0.03) | <0.007 |
| IPTW with IPCW# | −0.09 | (−0.17, −0.02) | 0.016 | −0.11 | (−0.18, −0.04) | <0.002 |
| Number of breastfeeding difficulties* | | | | | | |
| Unadjusted | 0.37 | (0.13, 0.61) | <0.002 | 0.50 | (0.28, 0.71) | <0.001 |
| IPTW‡ | 0.47 | (0.19, 0.74) | <0.001 | 0.50 | (0.24, 0.76) | <0.001 |
| IPTW with IPCW# | 0.42 | (0.15, 0.70) | <0.002 | 0.48 | (0.22, 0.74) | <0.001 |
| Any breastfeeding difficulties† | | | | | | |
| Unadjusted | 0.08 | (0.01, 0.16) | 0.028 | 0.24 | (0.17, 0.030) | <0.001 |
| IPTW‡ | 0.10 | (0.01, 0.19) | 0.027 | 0.23 | (0.15, 0.31) | <0.001 |
| IPTW with IPCW# | 0.08 | (−0.01, 0.17) | 0.065 | 0.22 | (0.14, 0.30) | <0.001 |

95% CI, 95% confidence interval; EPDS, Edinburgh Postnatal Depression Scale; IPCW, inverse probability of censoring weight; IPTW, inverse probability of treatment weight; p, p-value; PD, prevalence difference.

The unadjusted analysis included 833 women, the IPTW analysis included 803 women, and the IPTW with IPCW analysis included 751 women.

* The marginal structural model for EPDS and number of difficulties were modeled as linear Poisson, where point estimates correspond to differences in counts.

† The marginal structural model for probable depression and any breastfeeding difficulties were linear binomial. Point estimates for these results correspond to PDs. Probable depression was defined as an EPDS score of at least 13 at 14 weeks postpartum. Any breastfeeding difficulties was at least 1 self-reported difficulty of breastfeeding at week 10.

‡ IPTWs were conditional on age, education, marital status, previous children, prior miscarriage/abortion/stillbirth, not wanting additional children, prior experiences of domestic violence, home ownership, and flush toilet facility.

# IPCWs were conditional on age, education, marital status, previous children, prior miscarriage/abortion/stillbirth, not wanting additional children, prior experiences of domestic violence, home ownership, flush toilet facility, given their baby right after delivery, and intervention arm.

to SOC, with both Steps 1–9 and the modified Step 10 associated with a decrease in the EPDS score. We also showed that this association with depressive symptoms at 14 weeks was mediated by breastfeeding difficulties, with no evidence of an association of the BFHI steps with depressive symptoms in the absence of breastfeeding difficulties. A potential interpretation of these results is that when no breastfeeding difficulties were present, there was little to no associated stress encountered, and so nothing to "treat." In contrast, when too many difficulties were present, the BFHI steps may have been insufficient to prevent the development of depressive symptoms.

To our knowledge, this is the first study to look at the impact of the implementation of the 10 steps to successful breastfeeding that form the basis of BFHI on PPD symptoms. BFHI interventions, increase early initiation (within 1 hour of birth) of and prolonged exclusive breastfeeding [13], and thus are likely to be associated with increased levels of oxytocin among mothers. Higher levels of oxytocin are associated with lower reported breastfeeding difficulties and improved maternal mood [14,15]. However, our finding of no association between BFHI steps and depressive symptoms when breastfeeding difficulties was not present suggests that at least in settings like DRC where virtually all women initiate and continue breastfeeding for at least 6 months, other mechanisms must be at play. Social support and stress management skills

**Table 5. Estimated controlled direct associations between BFHI mediated through any difficulty breastfeeding at week 10.**

| | Steps 1–9 | | | Steps 1–10 | | |
|---|---|---|---|---|---|---|
| | Point estimate | 95% CL | p | Point estimate | 95% CL | p |
| EPDS score* | | | | | | |
| Mediation IPW‡ | | | | | | |
| No difficulty | 1.91 | (−6.60, 10.43) | 0.660 | 1.69 | (−1.17, 5.10) | 0.333 |
| At least 1 difficulty | −3.53 | (−4.56, −2.50) | <0.001 | −4.94 | (−5.95, −3.93) | <0.001 |
| With IPTW# | | | | | | |
| No difficulty | 4.77 | (−6.26, 15.80) | 0.396 | 1.61 | (−1.13, 4.34) | 0.249 |
| At least 1 difficulty | −3.45 | (−4.56, −2.33) | <0.001 | −5.12 | (−6.21, −4.03) | <0.001 |
| With IPTW and IPCW | | | | | | |
| No difficulty | 2.92 | (−7.04, 12.89) | 0.565 | 1.46 | (−1.32, 4.23) | 0.303 |
| At least 1 difficulty | −3.65 | (−4.80, −2.50) | <0.001 | −5.21 | (−6.33, −4.10) | <0.001 |
| Probable depression† | | | | | | |
| Mediation IPW‡ | | | | | | |
| No difficulty | 0.16 | (−0.23, 0.55) | 0.431 | 0.03 | (−0.20, 0.26) | 0.791 |
| At least 1 difficulty | −0.14 | (−0.21, −0.06) | 0.001 | −0.15 | (−0.23, −0.08) | <0.001 |
| With IPTW# | | | | | | |
| No difficulty | 0.29 | (−0.22, 0.79) | 0.263 | −0.03 | (−0.19, 0.13) | 0.735 |
| At least 1 difficulty | −0.14 | (−0.24, −0.04) | 0.004 | −0.15 | (−0.25, −0.06) | 0.002 |
| With IPTW and IPCW** | | | | | | |
| No difficulty | 0.21 | (−0.24, 0.66) | 0.365 | −0.03 | (−0.19, 0.13) | 0.735 |
| At least 1 difficulty | −0.15 | (−0.25, −0.06) | 0.001 | −0.16 | (−0.25, −0.06) | 0.001 |

95% CL, 95% confidence level; BFHI, Baby-Friendly Hospital Initiative; EPDS, Edinburgh Postnatal Depression Scale; IPCW, inverse probability of censoring weight; IPTW, inverse probability of treatment weight; IPW, inverse probability weight.

Any breastfeeding difficulties was at least 1 self-reported difficulty of breastfeeding at week 10.

* The marginal structural model for EPDS was modeled as linear Poisson, where point estimates correspond to differences in counts.

† The marginal structural model for Probable depression was linear binomial. Point estimates for these results correspond to PDs. Probable depression was defined as an EPDS score of at least 13 at 14 weeks postpartum.

‡ Mediation IPW used only the IPW weights calculated for any difficulties breastfeeding. Mediation IPWs were conditional on age, education, marital status, previous children, prior miscarriage / abortion / stillbirth, not wanting additional children, prior experiences of domestic violence, home ownership, flush toilet facility, given their baby right after delivery, exclusive breastfeeding up to week 10, and intervention arm.

# IPTW is IPW for mediation multiplied by IPTW. IPTWs for the intervention group were conditional on age, education, marital status, previous children, prior miscarriage / abortion / stillbirth, not wanting additional children, prior experiences of domestic violence, home ownership, and flush toilet facility.

** With IPTW and IPCW includes mediation IPW, IPTW, and IPCW. IPCWs were conditional on age, education, marital status, previous children, prior miscarriage / abortion / stillbirth, not wanting additional children, prior experiences of domestic violence, home ownership, flush toilet facility, given their baby right after delivery, intervention arms, and difficulties breastfeeding.

reducing depressive symptoms are both general phenomena. BFHI helps identify stressors, reframe them in a less stressful way, teach how to manage the stressors, and provide encouragement and social support to motivate people to effectively deal with stressors [5,6]. The importance of the identification of stressors is seen in the increased number of difficulties among women in the Steps 1–9 and Steps 1–10 arms. The increased identification of difficulties and the controlled direct association results indicate that the total effect of BFHI operates through the recognition of these stressors and coping with them.

PPD has serious impacts on the woman, her partner, and her baby. Women with PPD experience lower quality of life and an inability to care of themselves, their partner, and infants [28–30]. Consequently, infants of mothers with PPD are at increased risk of chronic and acute illness, death, and poorer physical, social, emotional, and intellectual development [31],

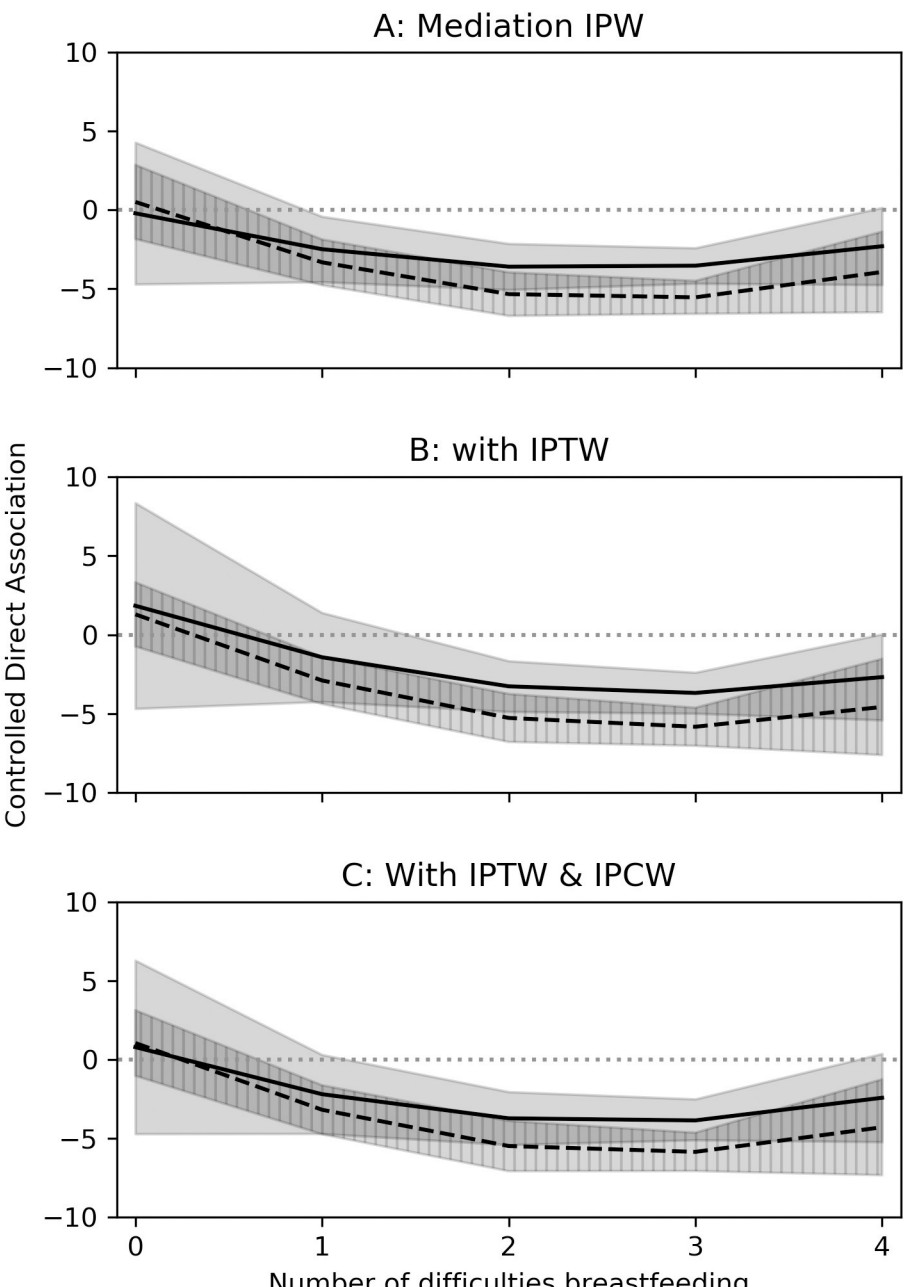

**Fig 2. Estimated controlled direct associations between BFHI and EPDS score at 14 weeks postpartum mediated through number of difficulties breastfeeding at week 10.** Solid line and solid shaded area indicated Steps 1–9 point estimate and CIs, respectively. Dashed line and hatched region indicated Steps 1–10 point estimate and CIs, respectively. BFHI, Baby-Friendly Hospital Initiative; CI, confidence interval; EPDS, Edinburgh Postnatal Depression Scale; IPW, inverse probability weight; IPCW, inverse probability of censoring weight; IPTW, inverse probability of treatment weight. The marginal structural model for EPDS and number of difficulties modeled was modeled as linear Poisson, where point estimates correspond to differences in counts.

resulting in lasting psychiatric and cognitive effects [32]. Our finding that the BFHI was associated with reduced depressive symptoms independently through a different path than that associated with increased breastfeeding provides additional evidence to support renewed efforts to scale it up, particularly in low-resource settings where access to mental health services is

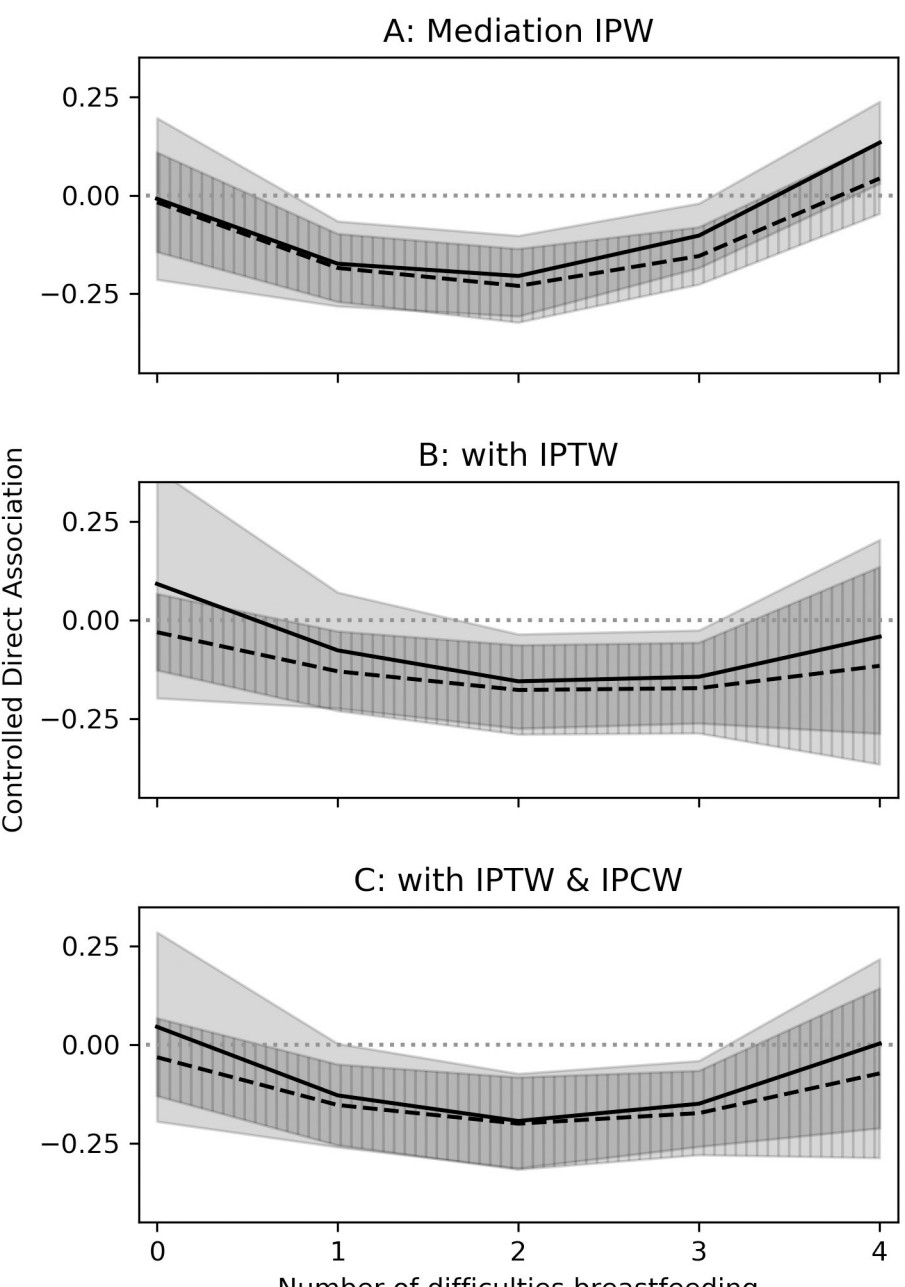

**Fig 3. Estimated controlled direct associations between BFHI and probable depression at 14 weeks postpartum mediated through number of difficulties breastfeeding at week 10.** Solid line and solid shaded area indicated Steps 1–9 point estimate and CIs, respectively. Dashed line and hatched region indicated Steps 1–10 point estimate and CIs, respectively. BFHI, Baby-Friendly Hospital Initiative; CI, confidence interval; IPW, inverse probability weight; IPCW, inverse probability of censoring weight; IPTW, inverse probability of treatment weight. The model for number of difficulties modeled was modeled as linear Poisson. The marginal structural model for probable depression was linear binomial, with point estimates corresponding to PDs. Probable depression was defined as an EPDS score of at least 13 at 14 weeks postpartum.

acutely limited. Coupled with the proven efficacy of the BFHI steps in increasing rates of exclusive breastfeeding, even partial implementation (Steps 1–9) is likely to have wide-reaching and long-lasting effects, both physical and psychiatric.

This study had a number of limitations. The small number of clusters precluded accounting for the clustering in the variance. Therefore, results may be optimistic in terms of precision. In addition, PPD was not a registered trial outcome nor was it included in the funded proposal (see S1 Protocol) as a secondary outcome. Further, some enrolled participants were lost to follow-up, thus a potential for selection bias. However, the consistency of the results from our unadjusted analyses, analyses adjusted for baseline characteristics and for baseline characteristics and censoring, and the magnitude of the estimates suggests that the observed association cannot all be explained by residual confounding or other potential bias.

With only 1 assessment of depressive symptoms (at week 14), we were unable to consider time-varying effects. These include the rate of reduction in symptoms, persistence of associations after the cessation of breastfeeding, and whether breastfeeding difficulties mediate the relationship at other time points. Most psychotherapeutic interventions are generally conducted for 12 to 20 weeks [33], suggesting that the timing of the EPDS administration here was well suited to detecting differences in depressive symptoms. However, our mediation results suggest that benefits will occur potentially as soon as breastfeeding difficulties are encountered, with the small caveat that the first few difficulties will still be relatively more stressful than later difficulties due to their novelty. With respect to persistence of associations, the mediated association through breastfeeding difficulties will of course no longer occur after the cessation of breastfeeding. However, a lighter symptom load will make future stressors easier to manage, reducing the risk of PPD symptoms. Whether the protective associations continue for a matter of weeks or a matter of months is unknown. Similarly, only 1 assessment of breastfeeding difficulties (at week 10) was considered, even though this information was collected at each of the study visits (6, 10, 14, 18, and 24 weeks postpartum). In addition to the fact that earlier difficulties might be relatively more stressful, the increasing protective benefit of BFHI on depressive symptoms with increasing difficulties found in our mediation analysis suggests a potential cumulative effect of the number difficulties over time including a differential impact of transient versus persistent difficulties. Breastfeeding assessments at 6 and 10 weeks can be incorporated into the mediation analysis for multiple mediators [34]. Such relationships might fruitfully be explored in future analyses.

In our analysis, we were unable to explore mechanisms by which BFHI steps exert their influence on postpartum depressive symptoms (i.e., social support quality and quantity, breastfeeding knowledge, and perceived stress of breastfeeding). As such, we are unable to identify what aspects of BFHI steps can be further improved to reduce PPD as well as increase exclusive breastfeeding. Similarly, it is unclear what approaches are best for improving social support (e.g., offering compassion and encouragement versus listening to issues) and breastfeeding knowledge (e.g., maximizing knowledge of major issues versus maximizing knowledge of all issues) without resulting in a decrease in exclusive breastfeeding prevalence as observed for Steps 1–10 in the original trial [16]. It is important for future research to address these questions to provide more actionable information.

For the identification of the average total association and the controlled direct association, we relied on several assumptions. We assumed exchangeability, causal consistency, positivity, and correctly specified parametric models [35]. For exchangeability, we assumed that there is no residual confounding of the treatment–outcome relationship and no residual confounding of the mediator–outcome relationship. Here, our IPW models controlled for common risk factors for depressive symptoms but were unable to directly control for preexisting depression symptoms. Prior symptom severity could affect treatment response, as well as the number and severity of breastfeeding difficulties. We were additionally unable to control for preexisting breastfeeding knowledge, beliefs, and social support, each of which could confound the mediator–outcome estimate. Causal consistency is the assumption that our treatment and mediator

are sufficiently well defined such that any additional variations of the treatment do not affect the outcome. We did not distinguish between types of breastfeeding difficulties. Some difficulties are clearly more severe and frustrating than others (e.g., pain versus leak), and the associations we discuss here may not apply equally to more stressful difficulties. Finally, positivity is the assumption that there are treated individuals at every level of confounders. Parametric regression models, and the corresponding assumption of no model misspecification, are necessary to weaken the positivity assumption for the estimation of IPW in the presence of high-dimensional data.

We further assumed no measurement error and non-informative missing data. Measurement error is a concern for 3 variables in particular: SES, difficulties breastfeeding, and EPDS scores. SES was measured through 2 durable goods. Difficulties with breastfeeding were more commonly reported in the Steps 1–9 and Steps 1–10 groups, which likely indicates a greater awareness of the difficulties and a willingness to report them, rather than an actual difference in the prevalence of difficulties breastfeeding. As indicated by our sensitivity analysis, the controlled direct association for no breastfeeding difficulties for Steps 1–9 should be interpreted cautiously. While we assume that other data are missing completely at random, IPCW results instead assume that missing depression data are non-informative conditional on demographics and clinic experience.

## Conclusions

In conclusion, we have shown that the implementation of the BFHI steps is associated with a decrease in depression symptoms at 14 weeks postpartum. The findings also suggest that this association is mediated by difficulties with breastfeeding. These results were robust across analytical approaches and different definitions of breastfeeding difficulties and probable depression. Given the numerous health consequences of PPD, the BFHI may then provide a general benefit to both maternal and infant health owing to the simple routine discussions about breastfeeding that the BFHI mandates, with this effect occurring in addition to those afforded by exclusive breastfeeding.

## Supporting information

**S1 Protocol. This is the funded proposal.**
(DOCX)

**S1 IPW Calculation. This file contains details on the calculation of weight used to implement the IPW. IPW, inverse probability weight.**
(DOCX)

**S1 STROBE Checklist. The completed STROBE Checklist. STROBE, Strengthening the Reporting of Observational Studies in Epidemiology.**
(DOC)

**S1 Sensitivity analyses. This file contains results of the sensitivity analyses.**
(DOCX)

**S1 Analytic Code (SAS). This is the SAS code used to produce the results.**
(SAS)

**S1 Data. This is the dataset used to produce the results.**
(XLSX)

## Acknowledgments

We thank the study mothers and infants for their participation and time and personnel of the participating clinics and study staff for their time and effort.

## Author Contributions

**Conceptualization:** Robert A. Agler, Marcel Yotebieng.

**Data curation:** Robert A. Agler, Paul N. Zivich, Bienvenu Kawende, Marcel Yotebieng.

**Formal analysis:** Robert A. Agler, Paul N. Zivich.

**Funding acquisition:** Frieda Behets, Marcel Yotebieng.

**Investigation:** Bienvenu Kawende, Frieda Behets, Marcel Yotebieng.

**Project administration:** Marcel Yotebieng.

**Supervision:** Bienvenu Kawende, Frieda Behets, Marcel Yotebieng.

**Writing – original draft:** Robert A. Agler, Marcel Yotebieng.

**Writing – review & editing:** Robert A. Agler, Paul N. Zivich, Bienvenu Kawende, Frieda Behets, Marcel Yotebieng.

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
