## [Editor Report · Decision Letter 0]

3 Feb 2020

Dear Dr Yotebieng, 

Thank you for submitting your manuscript entitled "Implementation of Ten Steps to Successful Breastfeeding Program Reduces Postpartum Depressive Symptoms:  a cluster-randomized controlled trial" for consideration by PLOS Medicine.

Your manuscript has now been evaluated by the PLOS Medicine editorial staff and I am writing to let you know that we would like to send your submission out for external peer review.

Kind regards,

Helen Howard, for Clare Stone PhD 

Acting Editor-in-Chief

PLOS Medicine 

plosmedicine.org

---

## [Decision Letter · Decision Letter 1]

10 Jun 2020

Dear Dr. Yotebieng,

Thank you very much for submitting your manuscript "Implementation of Ten Steps to Successful Breastfeeding Program Reduces Postpartum Depressive Symptoms:  a cluster-randomized controlled trial" (PMEDICINE-D-20-00294R1) for consideration at PLOS Medicine. 

Your paper was evaluated by a senior editor and discussed among all the editors here. It was also sent to four independent reviewers, including a statistical reviewer. The reviews are appended at the bottom of this email and any accompanying reviewer attachments can be seen via the link below:

[LINK]

In light of these reviews, I am afraid that we will not be able to accept the manuscript for publication in the journal in its current form, but we would like to consider a revised version that addresses the reviewers' and editors' comments. Obviously we cannot make any decision about publication until we have seen the revised manuscript and your response, and we plan to seek re-review by one or more of the reviewers. 

We expect to receive your revised manuscript by Jul 01 2020 11:59PM. Please email us (plosmedicine@plos.org) if you have any questions or concerns.

We look forward to receiving your revised manuscript. 

Sincerely,

Caitlin Moyer, Ph.D.

Associate Editor 

PLOS Medicine

plosmedicine.org

1. Title: Please revise the title to reflect that these are not the primary trial outcomes. Please also include information on the study population, we suggest: “Implementation of Ten Steps to Successful Breastfeeding Program and Postpartum Depressive Symptoms: additional outcomes from a cluster-randomized controlled trial in Kinshasa, Democratic Republic of Congo”

2. Throughout the manuscript: Because this manuscript reports on non-registered outcomes from the trial, please remove the use of causal language such as “effect of” “was mediated by”

3.Prospective analysis plan: Your outcomes differ from the registered trial protocol (NCT01428232). Did your study have a prospective protocol or analysis plan? Please state this (either way) early in the Methods section.

4. Data availability statement: Thank you for your willingness to share your data. Please provide the supplemental information spreadsheet (S4 data) as a separate file (not Zipped with SAS file) and include a descriptive legend explaining the column headers (variables). 

5. Abstract: Please structure your abstract using the PLOS Medicine headings (Background, Methods and Findings, Conclusions).

6. Abstract: Please describe the population demographics, including number of participants and the years during which the study took place.

7. Abstract: Line 26 (and throughout the manuscript): Please remove all references to causality, as these appear to be the post-hoc (additional) outcomes rather than the planned registered trial outcomes.

8.Abstract: Line 29-31 (and throughout manuscript): Please quantify the main results with both 95% CIs and p values.

9.Abstract: In the last sentence of the Methods and Findings section, please describe the main limitation(s) of the study's methodology.

10.Abstract Conclusions: Please address the study implications without overreaching what can be concluded from the data; the phrase "In this study, we observed ..." may be useful. Your results do not address infant health, so please revise or remove the following sentence: “The BFHI steps may improve infant health due to increased exclusive breastfeeding and maternal health due to reduced stress associated with infant feeding.”

11.Author Summary: At this stage, we ask that you include a short, non-technical Author Summary of your research to make findings accessible to a wide audience that includes both scientists and non-scientists. The Author Summary should immediately follow the Abstract in your revised manuscript. This text is subject to editorial change and should be distinct from the scientific abstract. Please see our author guidelines for more information: https://journals.plos.org/plosmedicine/s/revising-your-manuscript#loc-author-summary

12. Methods: Line 104-105: Trial registry: This manuscript does not appear to describe the primary or secondary outcomes of the trial as registered. Please clarify and explain the discrepancy. If the outcomes were not prespecified in the protocol, please indicate that they were post hoc and explain why they were added. Post hoc comparisons should be presented as hypothesis generating rather than conclusive. Please include the study protocol document and analysis plan, with any amendments, as Supporting Information to be published with the manuscript if accepted.

13. Methods: Lines 111-118: Please specify whether informed consent was written or oral.

14.Methods: Line 125-126: Although published elsewhere, please briefly describe the SOC condition.

15.Methods: Please provide the EPDS as a supporting information file, including information on scoring.

16.Results: Line 207-208: Here, and throughout, please remove causal language and temper statements such as “Among both intervention groups, the effects of the 207 BFHI on depression were mediated

208 through breastfeeding difficulties.” 

17. Discussion: Please present and organize the Discussion as follows: a short, clear summary of the article's findings; what the study adds to existing research and where and why the results may differ from previous research; strengths and limitations of the study; implications and next steps for research, clinical practice, and/or public policy; one-paragraph conclusion.

18. Checklist: Please complete the CONSORT checklist, (please use the extension for cluster-randomized trials: https://www.equator-network.org/reporting-guidelines/consort-cluster/) and ensure that all components of CONSORT are present in the manuscript. Please ensure that the study is reported according to the CONSORT guideline, and include the completed CONSORT checklist as Supporting Information. When completing the checklist, please use section and paragraph numbers, rather than page numbers. 

Please add the following statement, or similar, to the Methods: "This study is reported as per the Consolidated Standards of Reporting Trials (CONSORT) guideline (S1 Checklist)."

19. Table 2: The groups appear to differ, for example in the measure “baby given right after birth”- please comment on whether this difference is a limitation or could contribute to results.

20. Table 4 and Table 5 (and relevant section of the Results section text): Please provide both the 95% CIs and p values for all analyses.

Comments from the reviewers:

Reviewer #1: See attachment

Michael Dewey

Reviewer #2: In this article, the authors describe a study examining the impact of the Baby-Friendly Hospital Initiative (BFHI) on postpartum depression at 14 weeks postpartum for a sample of mothers in Kinshasa, Democratic Republic of the Congo.

This study represents a novel area of study. While previous studies have found the BFHI improves breastfeeding initiation and duration, and effective breastfeeding lowers mothers' risk of depression, this is the first to show that breastfeeding support--especially when there are problems--lowers mothers' risk of depression. 

As enthusiastic as I am about this topic, there are several areas where some clarification is needed.

Introduction

The authors work with the premise that both breastfeeding difficulties and mental illness subject mothers to shame and stigmatization. True, but these are not the only factors at play here. And many would argue (include some very vocal social media groups) that BFHI, and breastfeeding support in general, causes shame and stimatization. How does the BFHI lessen the impact of these? I note that this is a secondary analysis. However, I'd to see the authors show a somewhat deeper understanding of the complex relationship between depression and breastfeeding. Simply hypothesizing about possible shame and stigmatization only scratches the surface of possible underlying mechanisms and these seems to be the main mechanisms offered as an explanation for these findings.

Rather than something like shame as the major factor, there is likely a physiological mechanism underlying these results. For example, social support increases oxytocin, which lowers risk of depression (by downregulating the stress system). This helps with both breastfeeding and depression. In contrast, breastfeeding pain suppresses oxytocin, increasing both breastfeeding problems and depression risk. Skin-to-skin contact increases oxytocin and directly reduces stress. None of the steps does anything regarding shame and stigmatization. Nor do they directly address "common postpartum stressors" (discussion section), especially when implemented in the hospital. Many of those "common stressors" have nothing to do with breastfeeding and are therefore not addressed by BFHI.

For the purposes of this manuscript, I'd like to see the authors reference the wider PPD literature, particularly with regard to breastfeeding. I'd also suggest examining research in psychoneuroimmunology. I'd suggest the work of Janice Kiecolt-Glaser, Lisa Christian, Kerstin Uvnas-Moberg, and Kathleen Kendall-Tackett.

Please note a significant typo on Table 2. The abbreviation for the Edinburgh Postnatal Depression Scale is EPDS (not EPSD).

Interesting that exclusive breastfeeding rates seemed to be lower in mothers who received the full BFHI rather than steps 1-9. Can the authors comment on that?

Also, how do the authors explain that the BFHI seemed to be related to MORE problems than SOC. That would be a somewhat disturbing result. Also curiously, the EPDS scores appeared to be higher in the BFHI groups (Steps 1-9) when there were no breastfeeding problems.

Something else I'd like the authors to address is that what the mothers were identifying as problems all seem to be related to having "too much" milk. The problems that plague mothers in industrialized countries (sore nipples, not enough milk) are rare. This suggests that the results will not easily generalize to industrialized countries.

I've been around the BFHI for more than 20 years. It is not designed to help mothers with stress management (per Discussion section). I wouldn't even say that social support for the mother is the goal (although that happens indirectly). As the name implies, it's BABY-friendly. It's designed to set mothers and babies up for successful breastfeeding, partly by allowing natural processes to take place.

I found the second paragraph of the Discussion to be somewhat horrifying. OF COURSE, BFHI is not set up to deal with any postpartum mental illness, but especially severe illness. Heaven forbid! I've been training hospitals and baby-friendly providers for 25 years on how to recognize the symptoms of depression so they can be addressed, but not by breastfeeding supporters unless they are also mental health providers. But I've never thought that simply being in a progam like this was enough. And we should not represent it as such.

I also don't like paragraph 3. The risks of depression on mom and baby sound somewhat trite to me. How about including that PPD causes misery for the woman, her partner, and her baby. It impacts her interaction style with her infant, and that's what causes the problems downstream for babies (particularly because it can lead to insecure attachments). In the face of such serious problems, talking about a mother's return to her pre-pregnancy weight seems incredibly trivial. It's the type of comment that tends to rankle the mothers I've interviewed in my various studies. They often comment on "clueless" HCPs and it is one more reason why they stay silent. And how important is that issue in the DRC? Seems a bit like a first-world problem to me.

I note a number of really old references in the Reference section. Quite a few from the 1990s and some not specific to PPD. Some are not even related to depression. I'd urge the authors to update their understanding of PPD particularly as it relates to breastfeeding.

One final point. The EPDS, the primary measure of depression in this study, is actually a screening tool, not something that should be used to diagnose. I know that a lot of people do use it for a lot things it was never intended to be, but the authors need to acknowledge this as a limitation. In the current manuscript, that is not even mentioned. They also did not indicate how they scored it. What was the cutoff score they used to define depressed vs. non-depressed? Or did they use the EPDS as a continuous variable? Why did they choose that cutoff or decide to score it as a continuous variable? The authors need to provide more details here as well.

Reviewer #3: PLOS Review

PMEDICINE-D-20-00294R1

Implementation of Ten Steps to Successful Breastfeeding Program Reduces Postpartum Depressive Symptoms: a cluster-randomized controlled trial

This manuscript is an important addition to the evaluation of BFHI implementation and the impact on maternal health. Postpartum depression is a complex issues and support is an important part of the treatment strategy. The strengths of this paper include the quasi-experimental design. It is appropriate to randomize the intervention by clinic rather than by individual. The measure of postpartum depression has been validated. The statistical analysis is well described.

There are a few major issues that have not been covered by this manuscript.

* Study design implementation. 

o BFHI treatment fidelity at intervention sites. Were all of the steps implemented consistently during the evaluation period? The study assumes that the Steps were implemented and the only measure in this study of study participant exposure to the steps what the one question about being given the baby right after birth or Step 4 but studies have shown that Step 6 "Giving newborn infants no food or drink other than breastmilk, unless medically indicated, and not accepting free or low-cost breastmilk substitutes, feeding bottles, or teats" being very important to exclusive breastfeeding at discharge.

o Comparability of Sites: It would be helpful to have a description of the six sites and by important descriptive (operational descriptors geographic setting, size of site, size of maternity staff, number of deliveries per year).

* Sample Selection

o Inclusion and exclusion criteria are not discussed in the paper. Were preterm babies included? Were mothers and babies with medical conditions included? 

* Measurements:

o Maternal History of depression was not included. Mothers who have a history of depression before becoming pregnant or during pregnancy are much more likely to score high on the postpartum depression inventory. These mothers need to be analyzed separately or excluded from analysis.

o Mother's experience of BFHI steps Page 4, Ln 137-138. As mentioned above, only one step is assessed and does not reflect the entire BFHI experience. Mothers can report on Steps 4-10. You may not have been able to measure all of these steps but it should be acknowledged as a limitation in your discussion.

* Longitudinal data:

o Page 4 ln 117 "Women were enrolled within two days of delivery and follow-up at 6, 10, 14, 18, and 24-weeks postpartum." Indicates longitudinal data collected on you population. Although postpartum depression is only collected at week 14, is breastfeeding exclusivity and breastfeeding difficulties collected at multiple times? If so, timing of the difficulties (2 days, 6 weeks vs 10 or 14 weeks) would be important to investigate. Were earlier difficulties mediators of depression or were later difficulties or no difference. Most breastfeeding difficulties which impact exclusive breastfeeding happen early.

* Results

o Table 2: Characteristics of women at study enrollment by intervention group (n=975). Two results were not discussed in the results or discussion sections but seem to be very relevant to this study. The first was Any breastfeeding difficulties, Step 1-10 sites participants reported 91% compared with the SOC 67% and Steps 1-9 sites 75%. Sites 1-10 may have identified more breastfeeding difficulties as a result of attending support sessions after discharge. The other result not discussed was Exclusive breastfeeding up to week 10 SOC site participants reported 117 (38%), the Steps 1-9 - 237 (65%) and Steps 1-10 - 178 (58%). It is interesting that Steps 1-9 participants had higher rates of exclusive breastfeeding than the Steps 1-10 site participants. This finding in conjunction with the breastfeeding difficulties findings need to be explored more in discussion as they may related to the results.

o In Table 3: Reported breastfeeding difficulties at ten weeks post-partum (n=851), the most frequently cited breastfeeding difficulty for all three groups is Baby nursed too often with SOC-157 (58%), Steps 1-9-164 (54%) and Steps 1-10-161 (59%). This difficulty has not be a prominent difficulty in the literature and seems to be high in all three groups. At 14 weeks if might reflect a growth spurt and misunderstanding of baby developmental cues and growth needs by the mother which should be provided by a support group in the Steps 1-10 group. However even though breastfeeding difficulties are enumerated and used as a mediator, they are not discussed in the results nor the discussion. This is a major omission in this paper. Breastfeeding difficulties may provide information for future interventions on support of mothers. Depression is a multifaceted and multisensory concern. 'Nursing too often' may be a sign of an overwhelmed mother or a signal for outside support for other activities so that mother can provide the nutrition to the baby without the distraction or distress with not fulfilling other family activities.

Reviewer #4: An important paper in terms of assessing if BFHI with or without the community support group contact reduces maternal depression.

There are however some things that the authors should consider stating upfront. Since paper and findings are based on a "post-hoc" analyses of a trial data set, it should be stated clearly. 

Title: Since this paper reports on objectives that were never part of the original trial, not an outcome associated with the original trial, the title should not have RCT mentioned. It is misleading

Methods: 

Intervention description: The authors should clearly state and describe what the "tenth step" of the BFHI entailed? What was the frequency of the community support groups? Who led these? Were these peers based or supervised in any way? What was the content of these groups. 

Control Condition description should be done. at the moment not enough details given 

Recruitment: There seems to be a clear bias in recruitment - based on if the women wanted or not to be part of the BFHI. This is not random at all. The authors should comment on this. 

Analyses: Authors do not mention adjusting fro clustering, since this was a cRCT. A statistical review should be done to help authors in reporting. There seems to be some clear imbalances like domestic violence, education etc. Authors should state what was there approach, did they adjust for it? Also no mention about ITT analysis? If not ITT based analysis then why? 

Discussion: Should report or mention what the results were of the original trial? Did BFHI improve early initiation and EBF rates. This will help set the context for the findings of this paper. Since one of the outcomes the authors report is the "meditational aspect of feeding issues". The discussion should highlighted the limitations eg loss to follow up, baseline imbalances or issues with recruitment bias etc

[LINK]

---

## [Decision Letter · Decision Letter 2]

1 Sep 2020

Dear Dr. Yotebieng,

Thank you very much for submitting your manuscript "Implementation of Ten Steps to Successful Breastfeeding Program Reduces Postpartum Depressive Symptoms:  a cluster-randomized controlled trial" (PMEDICINE-D-20-00294R2) for consideration at PLOS Medicine. 

Your paper was evaluated by a senior editor and discussed among all the editors here. It was also discussed with an academic editor with relevant expertise, and sent to three independent reviewers, including a statistical reviewer. The reviews are appended at the bottom of this email and any accompanying reviewer attachments can be seen via the link below:

[LINK]

In light of these reviews, I am afraid that we will not be able to accept the manuscript for publication in the journal in its current form, but we would like to consider a revised version that addresses the reviewers' and editors' comments. Obviously we cannot make any decision about publication until we have seen the revised manuscript and your response, and we plan to seek re-review by one or more of the reviewers. 

We expect to receive your revised manuscript by Sep 22 2020 11:59PM. Please email us (plosmedicine@plos.org) if you have any questions or concerns.

We look forward to receiving your revised manuscript. 

Sincerely,

Caitlin Moyer, Ph.D.

Associate Editor 

PLOS Medicine

plosmedicine.org

1. Comment from the Academic Editor: Discussion: The authors do have serial assessments of breastfeeding difficulties, so their comment beginning on page 13 line 363 ("With only one assessment of depressive symptoms (at week 14) we were unable to consider time-varying effects...and whether breastfeeding difficulties mediate the relationship at other time points.") strikes me as a little incomplete. They could have incorporated the additional assessments of breastfeeding difficulties, but in their response to R3 they specifically state that they chose not to: "Yes, we did collect data on breastfeeding difficulties at each time point. We agree that the timing of the difficulties would be important to investigate. But, we will reserve this for another manuscript." So I would recommend revising the wording of these sentences (lines 363-366) to be more explicit about the type of analysis that they could have done but chose not to do. As written, the sentences are vague and unclear.

2. Title: Thank you for revising your title. After further discussion, we feel that the following title is more appropriate: “Postpartum depressive symptoms following implementation of the ten steps to successful breastfeeding program in Kinshasa, Democratic Republic of Congo: a cohort study” Please use the updated title in the manuscript submission form when submitting your revised manuscript (it does not seem to have been updated for the revised version).

3. Prospective analysis plan: The proposal included as a supporting information file does not mention the primary outcomes of the paper (i.e. depressive symptoms). If you have a prospective analysis plan for these outcomes (from your funding proposal, IRB or other ethics committee submission, study protocol, or other planning document written before analyzing the data), please include the relevant prospectively written document with your revised manuscript as a Supporting Information file to be published alongside your study, and cite it in the Methods section. Please specify in the Methods either way whether the analysis of these outcomes was pre-planned, and also note any departures from or additions to the prospective analysis plan and the reasons (e.g., following up on an unexpected result, request of a reviewer).

4. Ethical approval: In the Ethics section of the Methods, please explicitly demonstrate that ethical approval was received for this study, and the institution that granted approval (i.e. not just for the registered trial outcomes, but specifically for the observational outcomes described here).

5. Financial disclosure: The statement “We thank the study mothers and infants for their participation and time; personnel of the participating clinics and study staff for their time and effort.” would be more appropriate for the acknowledgements section.

6. Response to Reviewer 3: “Comparability of Sites: It would be helpful to have a description of the six sites and by important descriptive (operational descriptors geographic setting, size of site, size of maternity staff, number of deliveries per year).” Because PLOS Medicine does not have word limits for the manuscript text, please do provide these details as requested by the reviewer, either in the Methods or included within the supporting information files.

7. Response to Reviewers: Point raised by Reviewer 3 regarding language pertaining to measures of depressive symptoms with EPDS. Please refer to “depressive symptoms” throughout the manuscript rather than “depression” (For example, at Lines 194, 206-207, 270, 273 and in the legend of Table 5). In particular, the sentence "Depression was defined as an EPDS score of at least 13 at 14 weeks post-partum" should be changed to read "Probable depression was defined as…”

8. It appears that your study outcomes are observational and therefore causality cannot be inferred. Please remove language that implies causality, such as Lines 312-314 of the Discussion. Please refer to associations, rather than “effects” of the intervention on the outcomes.

9. Abstract: Please combine the Methods and Findings sections into one section, “Methods and findings”.

10. Abstract: Please define the abbreviation “SOC” at first use.

11. Abstract: Please quantify this result, with 95% CIs and p values: “We found mediation by breastfeeding difficulties, with little-to-no effect of BFHI steps if no breastfeeding difficulties occurred in the population.”

12. Abstract: Conclusions: We suggest revising to read: “In this study, we observed that depressive symptoms were reduced in the groups implementing BFHI Steps 1-9 or 1-10 relative to the standard of care, with implementation of Steps 1-10 resulting in the largest decrease. This reduction was mediated through breastfeeding difficulties. PPD has a huge negative impact on the mother, her partner, and the baby, with long lasting consequences. This additional benefit of BFHI steps suggest that renewed effort to scale its implementation globally may be beneficial to mitigate the negative impacts of postpartum depression on the mother, her partner, and the baby.” or similar

13. Author Summary: Please use bullet points rather than paragraph style for this section; please see our author guidelines for more information: https://journals.plos.org/plosmedicine/s/revising-your-manuscript#loc-author-summary

14. Results: Line 251-252: In the methods, please provide some details of the nature of excluding participants interviewed by one of the interviewers.

15. Results: Lines 262-266: Please report results with 95% CIs and p values.

16. Results: LInes 269-272: Please report results with 95% CIs and p values for: “In the presence of difficulties, EPDS scores and depression prevalence were decreased in step 1-9 and steps 1-10 groups compared to SOC. When no breastfeeding difficulties occurred, there was no observed effect of steps 1-10 on EPDS scores or depression.”

17. Results: Line 272: Please revise to clarify this result: “Minor increases in EPDS scores and the prevalence of depression did not reach statistical significance for steps 1-9 when all mothers were held to have no breastfeeding difficulties at week 10 for both IPTW and IPTW with IPCW results. However, these results were imprecise as indicated by the confidence interval width and differences did not reach statistical significance.” or similar. 

18. Results: Sensitivity analyses (Line 304): please reference the specific supporting information files here. Please also revise these sentences to clarify your meaning: However, EPDS scores were lowered at no difficulties for steps 1-9 and the prevalence of depressive symptoms was slightly protective. This is in contrast to Table 5 where a increases were seen. This indicates that the controlled direct effect was not robust to the modeling of exclusive breastfeeding at 10-weeks, but the remainder of the results were.”

19. Discussion: Please remove this sentence, as it seems like an over generalization given your study findings: “Similar improvements in symptomatology can also be expected for other psychiatric conditions, such as mania, anxiety, and psychosis because most psychiatric disorders are aggravated by stress and are improved by appropriate stressor management skills and social support.”

20. Table 2: Please change “EPDS Depression” to “EPDS score >13” or similar.

21. Figure 1: Please indicate the nature of those participants(pairs) that left the study at each stage. Please indicate the “cluster” structure of the study design in the flow chart.

22. Figures 2 and 3, and Supporting information figures 2.1 and 2.2 (Within Supplementary File 3): Please provide X and Y axis labels.

23. Checklist: Given that this is an observational study, authors should provide a STROBE checklist instead of CONSORT. Please ensure that the study is reported according to the STROBE guideline, and include the completed STROBE checklist as Supporting Information. When completing the checklist, please use section and paragraph numbers, rather than page numbers. Please add the following statement, or similar, to the Methods: "This study is reported as per the Strengthening the Reporting of Observational Studies in Epidemiology (STROBE) guideline (S1 Checklist)."

24. List of Supporting Information Files: The order and naming of files in the list does not match the file names.

25. Supplemental Table 3.2: Please provide p values to accompany these results.

Comments from the reviewers:

Reviewer #1: The authors have addressed my points.

Michael Dewey

Reviewer #3: The authors provided answers to all reviewer comments and addressed most concerns satisfactorily. I recommend acceptance of this publications with the following minor revisions. 1) causal language should be changed to meet editor guidance and 2) since the EPDS does not diagnose depression, depressive symptoms should be used and depressive symptoms >13 should be used and not the diagnosis of depression. 

Reviewer #4: Authors have addressed the points raised

[LINK]

---

## [Editor Report · Decision Letter 3]

30 Sep 2020

Dear Dr. Yotebieng,

Thank you very much for submitting your revised manuscript "Postpartum depressive symptoms following implementation of the ten steps to successful breastfeeding program in Kinshasa, Democratic Republic of Congo: a cohort study" (PMEDICINE-D-20-00294R3) for consideration at PLOS Medicine. 

Your paper was evaluated by a senior editor and discussed among all the editors here. It was also discussed with an academic editor with relevant expertise. The comments are appended at the bottom of this email.

Given the remaining editorial issues, we will not be able to accept the manuscript for publication in the journal in its current form, but we would like to consider a revised version that addresses the editors' comments. In particular, please revise accordingly with the academic editor's comment pertaining to the multiple observations of breastfeeding difficulties (please see point 1 below). Obviously we cannot make any decision about publication until we have seen the revised manuscript and your response. 

In revising the manuscript for further consideration, your revisions should address the specific points made by the editors. Please also check the guidelines for revised papers at http://journals.plos.org/plosmedicine/s/revising-your-manuscript for any that apply to your paper. In your rebuttal letter you should indicate your response to the editors' comments, the changes you have made in the manuscript, and include either an excerpt of the revised text or the location (eg: page and line number) where each change can be found. Please submit a clean version of the paper as the main article file; a version with changes marked should be uploaded as a marked up manuscript.

We expect to receive your revised manuscript by Oct 07 2020 11:59PM. Please email us (plosmedicine@plos.org) if you have any questions or concerns.

We look forward to receiving your revised manuscript. 

Sincerely,

Caitlin Moyer, Ph.D.

Associate Editor 

PLOS Medicine

plosmedicine.org

1. Response to academic editor's point 1: Thanks for revising the text in response to the academic editor’s comment regarding only considering one assessment of breastfeeding difficulties at 10 weeks. However, we request that you please edit to clarify this text as follows: 

"Similarly, only one assessment of breastfeeding difficulties (at week 10) was considered, even though this information was collected at each of the study visits (6, 10, 14, 18, and 24 weeks postpartum). In addition to the fact that earlier difficulties might be relatively more stressful, the increasing protective benefit of BFHI on depressive symptoms with increasing difficulties found in our mediation analysis suggest a potential cumulative effect of the number of difficulties overtime including a differential impact of transient vs persistent difficulties. The breastfeeding assessments at 6 and 10 weeks could have been incorporated into the mediation analysis by [please describe here the kind of analysis that could have been done with these additional assessments and how they would have added to the analysis described in the present manuscript]. Such relationships might fruitfully be explored in future analyses." 

2. Abstract: Methods and Findings: Please provide some summary demographics on the women included in the study such as age, parity, exclusive breastfeeding status, and proportions reporting any breastfeeding difficulty.

3. Abstract: Methods and Findings: Line 39-40: Please revise to: “However, a limitation of the study is that the results are based on two hospitals randomized to each group.”

4. Abstract: Conclusions: Line 43: Please change “resulting in” to “associated with” and at Line 44, please remove the word “huge”

5. Author summary: What did the researchers do and find?: In the second bullet point, please change “resulting in” to “associated with”

6. Author summary: What do these findings mean?: Please summarize the findings of your study, such as “Our results suggest that implementation of the baby-friendly hospital initiative steps was associated with a reduction in depressive symptoms, and that breastfeeding difficulties play a role in this relationship.”

7. Introduction: Line 127-128: Please revise to “Our objective was to quantify the association between implementing BFHI Steps 1-9 or Steps 1-10 and postpartum depressive symptoms.”

8. Methods: Please specify the significance level used (eg, P<0.05, two-sided) and the statistical test used to derive a p value.

9. Results: Lines 269-270: Please remove the word “substantively” from this sentence, and replace with “significantly” if statistical significance is meant (and please provide the 95% CIs/p values associated with the results here) because “substantively” is subjective (for example, 67% and 75% prevalence of difficulty may not be very different).

10. Results: Lines 278-280: Please clarify this sentence to read “Additionally, the prevalence of any reported difficulties was observed to be higher for steps 1- 9, although this did not reach statistical significance (0.08; 95% CL: -0.01, 0.17; p=0.06), and was significantly higher steps 1-10 (0.22; 95% CL: 0.14, 0.30; p<0.01).”

11. Results: Lines 287-289: Please revise this sentence to avoid confusion (it isn’t clear what a minor increasing association is), and please present the point estimate with 95%CIs and p values in the text: “When no breastfeeding difficulties occurred, we did not find evidence of a significant association of either intervention with EPDS scores or probable depression (Table 5)” 

12. Results: Lines 289-291: In line with the above revision, the following two sentences can be removed: “However, these estimates are small and imprecise, as indicated by the width of the confidence intervals. Therefore, there is insufficient indication of increased EPDS scores for either intervention if no breastfeeding difficulties was held constant.”

13. Results: Lines 322-330: Please revise these sentences to reflect the fact that there were no significant increases in EPDS or probable depression with the zero reported difficulties breastfeeding, and as shown in table 3.1, there are also no significant changes in EPDS or probably depression under zero reported difficulties. “The controlled direct association for both EPDS scores and probable depression reversed directions under no breastfeeding difficulties (Supplemental Table 3.1). However, similar to the results for mediation for Steps 1-9 presented in Table 5, changes in EPDS scores and probable depression under no difficulties breastfeeding did not reach statistical significance, suggesting that there is no association between steps 1-9 and EPDS at no breastfeeding difficulties at week 10.”

14. Discussion: Lines 339-340: Please revise this to “...with no evidence of an association of the BFHI steps with depressive symptoms in the absence of breastfeeding difficulties.”

15. Discussion: Lines 366-369: Please revise this to reduce causal language: “Our findings that the BFHI was associated with reduced depressive symptoms…”

16. Discussion: Lines 433-435: This sentence is confusing as there was no significant effect identified, please remove or clarify to: “As indicated by our analyses, we did not identify a significant controlled direct association for no breastfeeding difficulties for steps 1-9.”

17. Conclusions: Lines 439-440: Please revise to “In conclusion, we have shown that implementation of the BFHI steps is associated with a decrease in depression symptoms at 14-weeks postpartum.”

18. Table 2 and Table 3: Please make it clear in the legend or the Table headings that “Steps 1-9” and “Steps 1-10” refer to the BFHI (and please spell out the abbreviation). In Table 2, the comparison group is “Standard of Care” and in Table 3 this is “Control” - please use consistent terminology throughout.

19. S5 Data file: Please provide a summary tab, or similar to explain the variable names and cell values.

[LINK]

---

## [Editor Report · Decision Letter 4]

23 Oct 2020

Dear Dr. Yotebieng,

Thank you very much for re-submitting your manuscript "Postpartum depressive symptoms following implementation of the ten steps to successful breastfeeding program in Kinshasa, Democratic Republic of Congo: a cohort study" (PMEDICINE-D-20-00294R4) for review by PLOS Medicine.

I have discussed the paper with my colleagues and the academic editor. I am pleased to say that provided the remaining editorial and production issues are dealt with we are planning to accept the paper for publication in the journal.

[LINK]

In revising the manuscript for further consideration here, please ensure you address the specific points made by the editors. In your rebuttal letter you should indicate your response to the editors' comments and the changes you have made in the manuscript. Please submit a clean version of the paper as the main article file. A version with changes marked must also be uploaded as a marked up manuscript file.

We look forward to receiving the revised manuscript by Oct 30 2020 11:59PM. 

Sincerely,

Caitlin Moyer, Ph.D.

Associate Editor 

PLOS Medicine

plosmedicine.org

Requests from Editors:

1.Title: Please capitalize the “A” after the colon in “A cohort study”

2. Short title: Please revise to “Ten Steps to successful breastfeeding program and postpartum depressive symptoms” to avoid causal implications

3. Abstract: Background: Please clearly state the study’s objective or hypothesis as the final sentence of the Background section of the Abstract.

4. Abstract: Line 33: Please clarify that the difficulties should be “breastfeeding difficulties”

5. Abstract: Line 42: Please revise to: “there was no difference”

6. Abstract: Line 46-47: Please revise the first sentence of the Abstract conclusions to: "In conclusion, in this cohort, the implementation of the BFHI steps was associated with..."

7. Abstract: Conclusions: Line 48-49: Please clarify to “Specifically, the reduction in depressive symptoms was observed for women reporting breastfeeding difficulties.”

8. Author summary: What did the researchers do and find?: Please clarify the second bullet point, we suggest: “The reduction in depressive symptoms associated with implementation of BHFI Steps 1-9 or Steps 1-10 was observed for women reporting breastfeeding difficulties, but not for women reporting no difficulties.” or similar.

9. Methods: Line 143-144: Please reference the supporting information protocol document here “...the plan for this analysis was not included in the study protocol (S1 Protocol)” For the analyses that were not described in your protocol, please make sure that the Methods section transparently describes when the analyses were planned, and when/why any data-driven changes to analyses took place, including any additional analyses included in response to peer review comments. In particular, please note if the sensitivity analyses were pre-planned and if not, when they were added.

10. Methods Line 191-192: Please refer to the included proposal as a numbered file (e.g. S1 Protocol).

11.Methods: Line 256: Instead of “Supplement 2” please refer to the actual file name.

12. Results: LIne 281-282: Please revise to: “BFHI intervention implementations were associated with reduced EPDS scores and the prevalence of depressive symptoms when compared to SOC (Table 4).

13. Results: Line 283: Please revise to clarify:”...had a greater number of reported breastfeeding difficulties…”

14. Results: 293-293: Please give the point estimate with 95% CI and p values for this, as for the results presented above: “When no breastfeeding difficulties occurred, there was a positive but imprecise association of both interventions with EPDS scores or probable depression (Table 5).”

15. Results: LIne 297-298: “Mediation remained apparent when considering the number of breastfeeding difficulties instead (Figure 2, Figure 3).” Please present these results in the paragraph, or provide a table of these results (similar to Table 5, could be in the supporting information files) and refer to it.

16. Discussion: Line 341-343: Please revise to avoid causal language: “Our results provide evidence that the BFHI is associated with reduce postpartum depressive symptoms compared to SOC, with both Step 1-9 and the modified step 10 associated with a decrease in the EPDS score.”

17. Discussion: Line 346-348: Please revise to: “A potential interpretation of this results is that when no breastfeeding difficulties were present there was little to no associated stress encountered, and so nothing to “treat.” In contrast, when too many difficulties were present, the BFHI steps may have been insufficient to prevent the development of depressive symptoms.” or similar.

18. Discussion: line 353: Please place the reference in brackets before the comma, rather than after.

19. Discussion: line 381: Please refer to the specific file for your proposal (rather than referring to supplemental material).

20.Discussion: Thank you for the thorough discussion of your study’s limitations. If possible, please include a sentence such as "This study had a number of limitations..." or sub-header at the beginning of this section, to orient readers.

21. Discussion: Lines 447-449: Please revise to: “The findings also suggest that this association is mediated by difficulties with breastfeeding. These results were robust across analytical approaches and different definitions of breastfeeding difficulties and probable depression.”

22.Financial disclosure: Please remove this section from the body of the manuscript, and make sure all information is included and accurate in the Financial disclosure section of the manuscript submission form.

23. Reference List: Please check the formatting of the in-text citations (in square brackets before the punctuation mark and with no extraneous spaces) and all references in the list. Please double check the capitalization used in journal names (for example, The British journal of psychiatry in ref 22) and please use the "Vancouver" style for reference formatting, and see our website for other reference guidelines: https://journals.plos.org/plosmedicine/s/submission-guidelines#loc-references

24. Figure 1: If possible, please indicate on the flowchart at which weeks depressive symptoms/ breastfeeding difficulties were assessed.

[LINK]

---

## [Editor Report · Decision Letter 5]

16 Dec 2020

Dear Dr. Yotebieng,

I am writing concerning your manuscript submitted to PLOS Medicine, entitled “Postpartum depressive symptoms following implementation of the ten steps to successful breastfeeding program in Kinshasa, Democratic Republic of Congo: A cohort study.”

We have now completed our final technical checks and have approved your submission for publication. You will shortly receive a letter of formal acceptance from the editor.

Kind regards,

PLOS Medicine